# Mesoscopic modeling of hidden spiking neurons

**Shuqi Wang**[*], **Valentin Schmutz**[*], **Guillaume Bellec, Wulfram Gerstner**
Laboratory of Computational Neuroscience
École polytechnique fédérale de Lausanne (EPFL)
`first.lastname@epfl.ch`

## Abstract

Can we use spiking neural networks (SNN) as generative models of multi-neuronal recordings, while taking into account that most neurons are unobserved? Modeling the unobserved neurons with large pools of hidden spiking neurons leads to severely underconstrained problems that are hard to tackle with maximum likelihood estimation. In this work, we use coarse-graining and mean-field approximations to derive a bottom-up, neuronally-grounded latent variable model (neuLVM), where the activity of the unobserved neurons is reduced to a low-dimensional mesoscopic description. In contrast to previous latent variable models, neuLVM can be explicitly mapped to a recurrent, multi-population SNN, giving it a transparent biological interpretation. We show, on synthetic spike trains, that a few observed neurons are sufficient for neuLVM to perform efficient model inversion of large SNNs, in the sense that it can recover connectivity parameters, infer single-trial latent population activity, reproduce ongoing metastable dynamics, and generalize when subjected to perturbations mimicking optogenetic stimulation.

## 1 Introduction

The progress of large-scale electrophysiological recording techniques [1] begs the following question: can we reverse engineer the probed neural microcircuit from the recorded data? If so, should we try to design large spiking neural networks (SNN), representing the whole microcircuit, capable of generating the recorded spike trains? Such networks would constitute fine-grained mechanistic models and would make *in silico* experiments possible. However appealing this endeavor may appear, it faces a major obstacle – that of unobserved neurons. Indeed, despite the large number of neurons that can be simultaneously recorded, they add up to a tiny fraction of the total number of neurons involved in any given task [2], making the problem largely underdetermined. Training SNNs with large numbers of hidden neurons is challenging because a huge number of possible latent spike patterns result in the same recurrent input to the recorded neurons, making training algorithms nontrivial [3–6].

From the perspective of a single recorded neuron, the spike activity of all the other neurons can be reduced to a single causal variable – the total recurrent input (Figure 1A). Hence, we argue that fine-grained SNNs are not necessary to model the inputs from hidden neurons but can be replaced by a coarse-grained model of the sea of unobserved neurons. One possible coarse-graining scheme consists in clustering neurons into homogeneous populations with uniform intra- and inter-population connectivity. With the help of mean-field neuronal population equations [7–10], this approach enables the reduction of large SNNs to low-dimensional mesoscopic models composed of neuronal populations interacting with each other [11–13]. Clusters can reflect the presence of different cell-types [11, 14, 15] or groups of highly interconnected excitatory neurons [16–21]. From a computational point of view, coarse-grained SNNs offer biologically plausible implementations of

---

[*]Equal contributions.

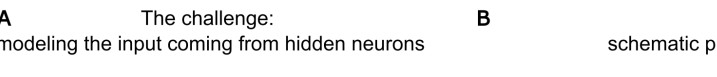

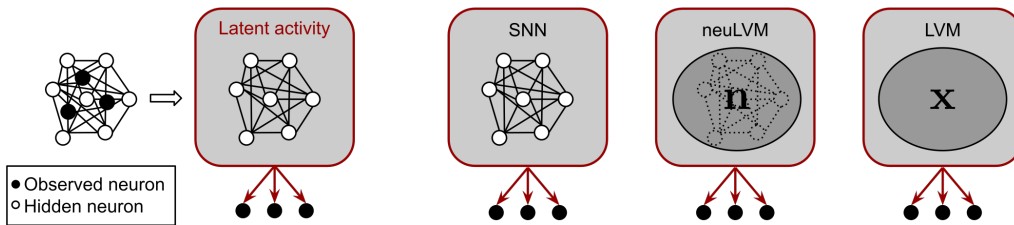

Figure 1: **Training SNNs with large numbers of hidden neurons: challenge and approaches.**
**(A)** The challenge of modeling the input to the observed neurons (black) coming from the hidden neurons (white) while only a small fraction of neurons is observed. **(B)** Modeling strategies for the input coming from the hidden neurons: SNNs (left) model the fine-grained spike trains of all hidden neurons; neuLVM (middle) uses a mesoscopic description of the population activity, clustering neurons into homogeneous populations; classic LVMs (right) model the latent activity with low-dimensional phenomenological variables (the link to SNNs is lost).

rate coding by ensembles of neurons [22, 23] and 'computation through neural population dynamics' [24].

In this paper, we show that, after clustering the unobserved neurons into several homogeneous populations, the finite-size neuronal population equation of Schwalger et al. [11] can be used to derive a neuronally-grounded latent variable model (neuLVM) where the activity of the unobserved neurons is summarized in a coarse-grained mesoscopic description. The hallmark of neuLVM is its direct correspondence to a multi-population SNN. As a result, both model parameters and latent variables have a transparent biological interpretation: the model is parametrized by single-neuron properties and synaptic connectivity; the latent variables are the summed spiking activities of the neuronal populations. Coarse-graining by clustering, therefore, turns an underdetermined problem – fitting an SNN with a large number of hidden neurons – into a tractable problem with interpretable solutions.

Switching metastable activity patterns that are not stimulus-locked have attracted a large amount of attention in systems neuroscience [25–27] for their putative role in decision-making [28], attention [29], and sensory processing [30]. Since generative SNN models of these metastable dynamics are available [11, 21, 31, 32], metastable networks constitute ready-to-use testbeds for bottom-up mechanistic latent variable models. Therefore, we propose metastable networks as a new benchmark for mechanistic latent variable models.

## 2 Relation to prior work

While many latent variable models (LVM), including Poisson Linear Dynamical Systems (PLDS) [33] and Switching Linear Dynamical Systems (SLDS) [34–39], have been designed for inferring low-dimensional population dynamics [40–53], their account of the population activity is a phenomenological one. By contrast, the LVM derived in this work is a true multiscale model, as latent population dynamics directly stems from neuronal dynamics.

Our method builds on René et al. [54], who showed that the mesoscopic model of Schwalger et al. [11] enables the inference of neuronal and connectivity parameters of multi-population SNNs via likelihood-based methods when the mesoscopic population activity is *fully observable*. Here, for the first time, we show that mesoscopic modeling can also be applied to unobserved neurons, relating LVMs to mean-field theories for populations of spiking neurons [7–12]. Our neuLVM approach towards unobserved neurons differs from the Generalized Linear Model (GLM) approach [55–57] (and recent extensions [58–60]), which either neglects unobserved neurons or replaces unobserved neurons by stimulus-locked inputs. Our approach also avoids microscopic simulations of hidden spiking neurons [3, 4, 6], which scale poorly with the number of hidden neurons.

# 3 Background: mesoscopic modeling of the population activity

Biophysical neuron models can be accurately approximated (neglecting the nonlinearity of dendritic integration) by simple spiking *point* neurons [61–63], which can be efficiently fitted to neural data [64–67]. Stochastic spiking neuron models where the neuron's memory of its own spike history is restricted to the time elapsed since its last spike (its *age*) are said to be of renewal-type.[2] Examples of renewal-type neurons include noisy leaky integrate-and-fire and 'Spike-Response Model 0' neurons [9, 10]. The dynamics of a homogeneous population of interacting renewal-type neurons can be described, in the mean-field limit, by an exact integral equation [7, 9, 10, 12, 69] (see [70–72] for rigorous proofs). In the case of homogeneous but finite populations, Schwalger et al. [11] derived a stochastic integral equation that provides a mesoscopic description (i.e. a description including finite-size fluctuations) of the population dynamics.

For clarity of exposition, in this section and the next, we focus on the case of a single homogeneous population with no external input. All the arguments presented below can be readily generalized to the case of multiple interacting populations with external input (Appendices A, B, C).

Let us consider a homogeneous SNN of $N$ recurrently connected renewal-type spiking neurons. For $T$ discrete time steps of length $\Delta t$, let $\mathbf{y} \in \{0, 1\}^{N \times T}$ (a $N \times T$ binary matrix) denote the spike trains generated by the $N$ neurons. The fact that the neurons are of renewal-type implies, by definition, that the probability for neuron $i$ to emit a spike at time $t$ can be written $p(y_t^i = 1 | \mathbf{y}_{1:t-1}, \Theta) = \rho_{\theta^i}^{\Delta t}(a^i, \sum_j J^{ij} \mathbf{y}_{1:t-1}^j)$ where $a^i$ is the age of neuron $i$ (i.e. the number of time steps elapsed since the last spike of neuron $i$), the $J^{ij}$ are the recurrent synaptic weights of the network, and $\theta^i$ are the parameters of neuron $i$. The sum $\sum_j J^{ij} \mathbf{y}_{1:t-1}^j$ represents the past input received by neuron $i$ in all time steps up to $t - 1$. The superscript $\Delta t$ of the function $\rho_{\theta^i}^{\Delta t}$ indicates that we consider here the discrete-time 'escape rate' of the neuron but the transition to continuous time is possible [10]. The explicit expression for $\rho_{\theta^i}^{\Delta t}$ in the case of leaky integrate-and-fire neurons with 'escape noise' (LIF) is given in Appendix A.

A crucial notion in this work is that of a 'homogeneous population'. The SNN described above forms a homogeneous population if all the recurrent synaptic weights are identical, that is, $J^{ij} = J/N$ (mean-field approximation), and if all the neurons share the same parameters, that is, $\theta^i = \theta$. In a homogeneous population, all the neurons share the same past input $J\mathbf{n}_{1:t-1}/N$, where $\mathbf{n}_{1:t-1} = (n_1, n_2, \ldots, n_{t-1})$ denotes the total number of spikes in the population in time steps $1, 2, \ldots, t - 1$ with $n_{t'} = \sum_{i=1}^N y_{t'}^i$ being the total number of spikes in the population at time $t'$. Then, for *any* neuron in the population, the probability to emit a spike at time $t$, given its age $a$, is

$$p_{t,a}^{\text{fire}} = \rho_\theta^{\Delta t}(a, J\mathbf{n}_{1:t-1}/N). \tag{1}$$

Importantly, Eq. (1) is independent of the identity of the neuron.

In a microscopic description of the spiking activity, the vector $\mathbf{y}_t$ depends nonlinearly on the past $\mathbf{y}_{1:t-1}$. A mesoscopic description aims to reduce the high-dimensional microscopic dynamics to a lower-dimensional dynamical system involving the population activity $n_t$ only (in the case of multiple interacting populations, $\mathbf{n}_t$ is a vector of dimension $K$ equal to the number of populations, Appendix A). While an exact reduction is not possible in general (neuron models being nonlinear), a close approximation in the form of a stochastic integral equation was proposed by Schwalger et al. [11]. In discrete time, the stochastic integral equation reads

$$n_t \sim \text{Binomial}\left(N, \ \bar{n}_t/N\right), \tag{2a}$$

$$\bar{n}_t = \left[\sum_{a \geq 1} p_{t,a}^{\text{fire}} S_{t,a} \, n_{t-a} + \underbrace{\Lambda_t \left(N - \sum_{a \geq 1} S_{t,a} \, n_{t-a}\right)}_{\text{'finite-size correction'}}\right]_+. \tag{2b}$$

The variable $\bar{n}_t$ can be interpreted as the expected number of neurons firing at time $t$. The survival $S_{t,a} = \prod_{s=0}^{a-1}(1 - p_{t-a+s,s}^{\text{fire}})$ is the probability for a neuron to stay silent between time $t - a$ and

---

[2]Traditional renewal theory in the mathematical literature [68] is restricted to stationary input whereas we use 'renewal-type' in the broader sense that also includes time-dependent input.

$t-1$. The finite-size correction term stabilizes the model by enforcing the approximate neuronal mass conservation $\sum_{a \geq 1} S_{t,a} n_{t-a} \approx N$ (see [13] for an in-depth mathematical discussion). The 'modulating factor' $\Lambda_t$ has an explicit expression [11, 13] in terms of $p^{\text{fire}}$, $S$ and $\mathbf{n}$ (indices are dropped here for simplicity, complete formulas are presented in Appendix A, as well as explanations on how to initialize Eq. (2)). Importantly, for a populations of *interacting* neurons, $p^{\text{fire}}$, $S$ and $\Lambda$ depend on $\mathbf{n}$, which makes the stochastic equation (2) highly nonlinear. While the mesoscopic model (2) is not mathematically exact, it provides an excellent approximation of the first and second-order statistics of the population activity [11], and is much more tractable than the exact 'field' equation [73, 74]. Also, the mesoscopic model (2) can be generalized to the case of non-renewal neurons with spike-frequency adaptation [11] and short-term synaptic plasticity [75].

Formally, Eq. (2) is reminiscent of the Point Process Generalized Linear Model (GLM) [55–57] for single neurons, with the notable difference that Eq. (2) contains additional nonlinearities beyond those of the GLM because $p^{\text{fire}}$, $S$ and $\Lambda$ all depend on $\mathbf{n}$ (Appendix A). Importantly, Equation (2) readily defines an expression for the probability $p(\mathbf{n}|\Theta)$ [54], where $\Theta = \{J, \theta\}$ denotes the model parameters. Thus, the mesoscopic model (2) allows us to avoid the intractable sum encountered if we naively try to derive $p(\mathbf{n}|\Theta)$ directly from the microscopic description (the intractable sum stems from the fact that the identity of neurons is lost in the observation $n_{t'}$ at each time step $t'$, Figure 1B).

## 4    Theoretical result: Neuronally-grounded latent variable model

In this section, we first recall why training SNN with large numbers of hidden neurons via the maximum likelihood estimator is computationally expensive. Then, we show that the mesoscopic description, Eq. (2), allows us to derive a tractable, neuronally-grounded latent variable model, which can be mapped to a multi-population SNN.

For the sake of simplicity, all the arguments are presented for a single homogeneous population, but the generalization to multiple interacting populations is straightforward (Appendices B and C). Let us assume that we observe, during $T$ time steps, the spike trains of $q$ simultaneously recorded neurons that are part of a homogeneous population of $N$ neurons, with $N > q$. We split the spike trains of the entire population $\mathbf{y} \in \{0,1\}^{N \times T}$ into the observed spike trains $\mathbf{y}^{\mathbf{o}}$ ($q$ neurons) and hidden spike trains $\mathbf{y}^{\mathbf{h}}$ ($N - q$ neurons). Even for a single population, it is difficult to infer the parameters $\Theta = \{J, \theta\}$ of the SNN from observation $\mathbf{y}^{\mathbf{o}}$ using the maximum likelihood estimator because, in the presence of hidden neurons, the likelihood $\mathcal{L}$ involves a marginalization over the latent spike trains $\mathbf{y}^{\mathbf{h}}$:

$$\mathcal{L} = p(\mathbf{y}^{\mathbf{o}}|\Theta) = \sum_{\mathbf{y}^{\mathbf{h}}} p(\mathbf{y}^{\mathbf{o}}, \mathbf{y}^{\mathbf{h}}|\Theta). \tag{3}$$

While different variants of the Expectation-Maximization (EM) algorithm [76] relying on sampling $\mathbf{y}^{\mathbf{h}}$ have been used to maximize the likelihood [3, 4, 6], these algorithms scale poorly with the number of hidden neurons.

Instead, we exploit the fact that, for a homogeneous population, the fine-grained knowledge of the latent activity $\mathbf{y}^{\mathbf{h}}$ is not necessary since all the observed neurons receive at time $t$ the same input $Jn_t$, where $n_t = \sum_{i=1}^{N} y_t^i$ is the population activity of Section 3. Hence, we rewrite the likelihood (3) as

$$\mathcal{L} = p(\mathbf{y}^{\mathbf{o}}|\Theta) = \sum_{\mathbf{n}} p(\mathbf{y}^{\mathbf{o}}, \mathbf{n}|\Theta), \tag{4a}$$

where the probability $p(\mathbf{y}^{\mathbf{o}}, \mathbf{n}|\Theta)$ factorizes in $T$ terms of the form

$$p(\mathbf{y}_t^{\mathbf{o}}, n_t | \mathbf{y}_{1:t-1}^{\mathbf{o}}, \mathbf{n}_{1:t-1}, \Theta) = \underbrace{p(\mathbf{y}_t^{\mathbf{o}} | \mathbf{y}_{1:t-1}^{\mathbf{o}}, \mathbf{n}_{1:t-1}, \Theta)}_{\text{given by neuron model, Eq. (1)}} \underbrace{p(n_t | \mathbf{n}_{1:t-1}, \Theta)}_{\text{approx. by meso. model, Eq. (2)}} . \tag{4b}$$

A comparison of Eqs. (3) and (4) shows that the high-dimensional latent activity $\mathbf{y}^{\mathbf{h}}$ has been reduced to a low-dimensional mesoscopic description. Importantly, the $q$ observed spike trains are conditionally independent given the population activity $\mathbf{n}$. While the conditional dependence structure implied by Eq. (4b) is typical of standard latent variable models of multi-neuronal recordings [33, 40, 77, 78], in our approach, the latent variable explicitly represents the population activity

of the generative SNN and the parameters of the model are identical to those of the SNN. As the latent population dynamics directly stems from neuronal dynamics, we call our LVM the neuronally-grounded latent variable model (neuLVM).

The nonlinearity and the non-Markovianity of Eq. (2) prevent us from using previous EM algorithms [33, 40, 77, 78]. Therefore, we fit the neuLVM via the Baum-Viterbi algorithm [79] (also known as Viterbi training or hard EM [80]), which alternates estimation (E) and maximization (M) step

**E-step**. $\widehat{\mathbf{n}}^n = \operatorname{argmax}_{\mathbf{n}} \log p(\mathbf{y}^{\mathbf{o}}, \mathbf{n} | \widehat{\Theta}^{n-1})$,

**M-step**. $\widehat{\Theta}^n = \operatorname{argmax}_{\Theta} \log p(\mathbf{y}^{\mathbf{o}}, \widehat{\mathbf{n}}^n | \Theta)$.

The estimated parameters $\widehat{\Theta}$ and the estimated latent population activity $\widehat{\mathbf{n}}$ are the result of many iterations of **E-step** and **M-step** (Appendix C). Note that the computational cost of this algorithm does not depend on the number of hidden neurons (it only depends on the number of populations).[3]

## 5  Experimental results

### 5.1  Single homogenous population: SNN with metastable cluster states

Although seemingly simple, homogeneous populations of leaky integrate-and-fire (LIF) neurons without external stimulation are SNNs with a rich repertoire of population dynamics, including asynchronous states, synchronous states, and cluster states [7, 9]. In a $m$-cluster state (with $m \geq 2$), the population activity oscillates at a frequency $m$ times higher than the average neuronal firing rate: a neuron spikes every $m$ cycle on average; conversely, approximately $N/m$ neurons fire in each cycle ($N$ being to the total number of neurons). Cluster states have therefore been described as 'higher harmonics' of the synchronous state (or 1-cluster state) [9, 81–83] where all neurons fire in synchrony.

In this set of experiments, we always consider the same network of 600 LIF neurons (Figure 2A), where only the connectivity parameter $J$ varies. When initialized at time 0 in the same unstable asynchronous state, the network can spontaneously settle in a $m$-cluster state, where $m$ depends on the recurrent connectivity parameter $J$ (Figure 2B): finite-size fluctuations break the symmetry of the asynchronous state and the population splits into $m$ groups of synchronized neurons. The cluster state to which the network converges can be read from the power spectrum of the neuronal spike trains (Figure 2B) (the fundamental frequency of the $m$-cluster state is approximately $m$ times lower than that of the 1-cluster state). Generating spike trains for 6 observed neurons (1% of the population), we tested whether neuLVM could recover the connectivity parameter $J$ (neuronal parameters $\theta$ were given), for different $J$'s in the 1-, 2-, and 3-cluster states range (Figure 2C, Table S3). The Pearson correlation between the learned $\widehat{J}$ and the true $J$ was 0.81 with p-value 2.8e−17, showing that, statistically, neuLVM could recover the connectivity parameter of the SNN.

To assess how well neuLVM can infer the latent population activity and how neuLVM compares with the methods assuming full observability (like René et al. [54]), we studied in detail a single trial showing a transition from a metastable 4-cluster state to a 3-cluster state (Figure 2D,E). To generate this trial, we chose $J = 60.32$ mV and initialized the network in the 4-cluster state. From the spike trains of only two neurons (red stars in Figure 2D), neuLVM could infer the ground truth population activity $\mathbf{n}^*$ during the 4-cluster state, and during the 3-cluster state, and could approximately detect the transition between the two states (Figure 2E). While the summed, smoothed spike trains missed two out of four population activity peaks in the 4-cluster state, and one out of three peaks in the 3-cluster state (purple curve in Figure 2E), the strong inductive biases contained in neuLVM enabled the inference of the 'missing' peaks (blue curve in Figure 2E). Finally, neuLVM and a method assuming full observability (equivalent to a naive application of René et al. [54]) were compared through their ability to recover the connectivity parameter $J$, for varying numbers of observed neurons (Figure 2F, Table S4). Since a naive application of the method of René et al. [54] does not take into account hidden neurons, it led, as expected, to wildly inaccurate estimates $\widehat{J}$ when the summed spike train was far from the ground truth population activity (which happened when the number of observed neurons was small, see Figure 2E for an example). In contrast, the neuLVM managed to recover the

---

[3]Our implementation of the algorithm is openly available at `https://github.com/EPFL-LCN/neuLVM`.

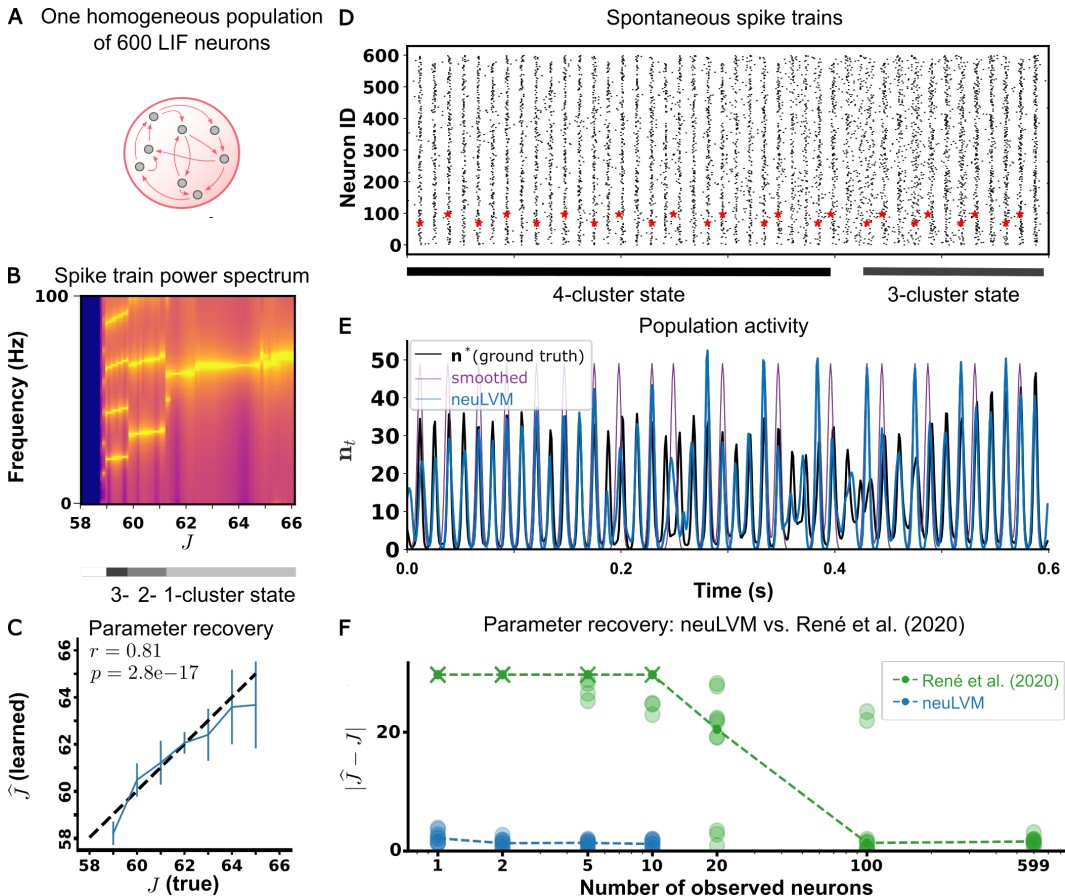

Figure 2: **Single-population SNN with metastable cluster states.** **(A)** Network architecture (for visualization purposes, only a few connections are drawn). **(B)** Spike train power spectrum for different choices of connectivity parameter $J$. All simulations start from the same unstable asynchronous state. The corresponding cluster states are indicated below. The blue region around $J = 58$ mV indicates the absence of activity. **(C)** Connectivity recovered by the neuLVM $\widehat{J}$ vs ground truth $J$. The neuLVM was fitted on one-second single-trial recordings of six neurons (1% of the population). For each ground truth $J$ value (seven in total), ten different trials were generated: bars indicate the standard deviations of the recovered $\widehat{J}$. The Pearson correlation coefficient between the recovered $\widehat{J}$ and $J$ is $r = 0.81$ and the associated p-value is $2.8\mathrm{e}{-17}$ (see Table S3). **(D)** Spike trains generated by the ground truth SNN for a trial showing a transition from a metastable 4-cluster state to a 3-cluster state. The spike trains of two randomly sampled neurons (red stars) formed the training data (for visualization purposes, only the first 0.6 second of the one-second trial is shown) on which neuLVM was fitted: **(E)** the inferred population activity $\widehat{\mathbf{n}}|\mathbf{y^o}$ is compared to the ground truth $\mathbf{n}^*$ and the summed, smoothed spike trains (Gaussian smoothing window with $\sigma = 1.4$ ms, Appendix D) of the two observed neurons. **(F)** Absolute difference between the recovered $\widehat{J}$ and the ground truth $J$ for the neuLVM algorithm and the method of René et al. (2020) for varying numbers of observed neurons. Using the same trial as in D, for each number of observed neurons, the two methods were tested on 10 different samples of observed neurons (see Table S4). The marker '$\times$' indicates that the difference $|\widehat{J} - J|$ is larger than 30 mV. The median samples are linked with dashed lines to show the trends.

connectivity parameter, thanks to the fact that the Baum-Viterbi algorithm of Section 4 also infers the population activity (see Figure 2E for an example).

## 5.2 Multiple populations: SNN with metastable point attractors

### 5.2.1 Latent population activity inference and reproduction of metastable dynamics

As a second benchmark, we tested neuLVM on synthetic data generated by three interacting populations with two populations of 400 excitatory LIF neurons and one population of 200 inhibitory neurons (Figure 3A). Recurrent connections between these population drive winner-take-all dynamics with finite-size fluctuations-induced switches of activity between the two excitatory populations [11, 84] – an example of 'itinerancy between attractor states' [25]. The population activities of this metastable, three-population SNN constitute the ground truth against which different models will be tested.

To build a spiking benchmark dataset, we randomly selected 9 neurons – 3 neurons from each of the three populations – and considered the spike trains of these neurons as the observed data. For simplicity, the correct partitioning of the 9 neurons into 3 groups is given since it can be reliably obtained by k-means clustering [85] using the van-Rossum-Distance [86] between spike trains. The complete dataset consists of 20 trials of 10 seconds. An example trial is shown in Figure 3B.

In contrast with the experiments of Section 5.1 where the neuronal parameters were given, here, neuronal and connectivity parameters are not given to neuLVM (see Appendix F). We compared the performance of neuLVM with other generative models of spiking data – PLDS [33], SLDS [39], and GLM [55-57] – on single trials of the spiking benchmark dataset in two ways: (i) we measured the Pearson correlation $r$ between the inferred latent population activity $\widehat{\mathbf{n}}|\mathbf{y^o}$ and the ground truth population activity $\mathbf{n}^*$ (Table 1 first column); (ii) we assessed how well could the fitted models reproduce metastable dynamics by counting the occurrences of stochastic switches in free simulations – or in other words, samples – of the fitted models (Table 1 second column). Tests (i) and (ii) on an example trial are shown in Figure 4.

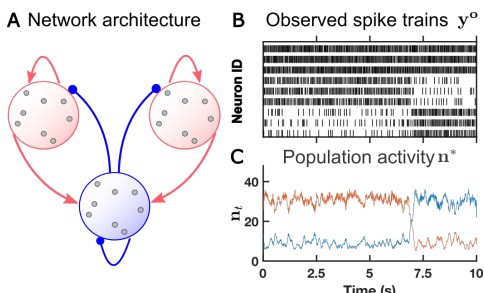

Figure 3: **Network architecture and example trial.** **(A)** Architecture of the three-population, metastable, winner-take-all SNN. **(B)** Example trial from the spiking benchmark dataset: 10 seconds recordings of 9 observed neurons (3 neurons from each of the three populations) and **(C)** corresponding ground truth latent population activity (for the two excitatory populations).

The Poisson Linear Dynamical Systems approach (PLDS, [33]) assumes that the recorded spikes can be explained by point processes driven by a latent linear dynamical system of low dimension. The Poisson Switching Linear Dynamical System (SLDS, [34-39]) extends PLDS by allowing the latent variables to switch randomly between two dynamical systems with distinct parameters. We should stress that, in PLDS and SLDS, the latent variables are *phenomenological* representations of neural population activity which have no direct link with the ground truth population activity $\mathbf{n}^*$. In order to still make test (i) possible for PLDS and SLDS, we will consider the best linear transformation of the inferred latent variables which minimizes the mean squared error with the ground truth population activity $\mathbf{n}^*$.

On test (i), neuLVM gave better estimates $\widehat{\mathbf{n}}|\mathbf{y^o}$ of the latent population activity $\mathbf{n}^*$ (Pearson correlation $r = 0.81$) than the best linear transformation of the latent activity inferred by PLDS and SLDS ($r = 0.69$ and $r = 0.73$ respectively) (Table 1 first column). The GLM approach cannot be included in test (i) since it ignores unobserved neurons. Interestingly, the example trial in Figure 4A shows the latent population activity $\widehat{\mathbf{n}}|\mathbf{y^o}$ inferred by neuLVM is smoother than the ground truth $\mathbf{n}^*$ before and after the switch (finite-size fluctuations are reduced) but $\widehat{\mathbf{n}}|\mathbf{y^o}$ and $\mathbf{n}^*$ closely match around the time of the switch. In contrast, fluctuations are exaggerated for PLDS and SLDS. The population activity

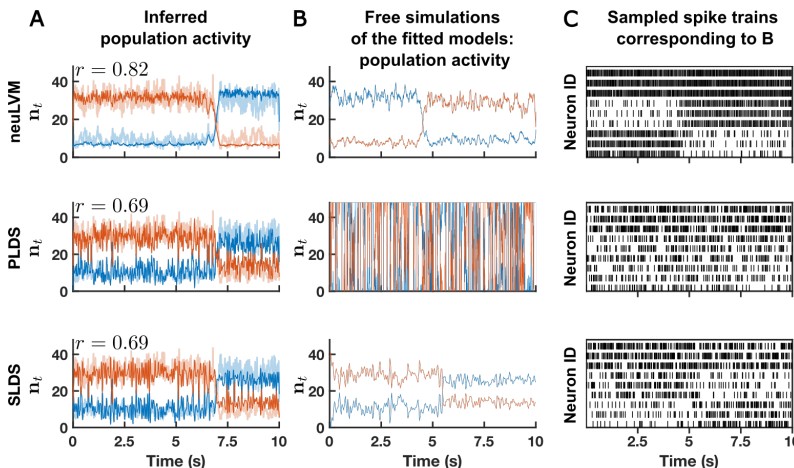

Figure 4: **Three-population SNN with metastable point attractors. (A)** Latent population activity of the two excitatory populations inferred by neuLVM / PLDS / SLDS for one example trial (the same as in Figure 3). The value $r$ is the Pearson correlation coefficient between the inferred $\widehat{\mathbf{n}}|\mathbf{y}^{\mathbf{o}}$ and the ground truth $\mathbf{n}^*$ population activities. **(B-C)** Examples of free simulations of the fitted neuLVM / PLDS / SLDS.

Table 1: Model performance summary (corresponding to Figure 4).

| Models | Pearson correlation $r$ between $\widehat{\mathbf{n}}\|\mathbf{y}^{\mathbf{o}}$ and $\mathbf{n}^*$ | Number of switches during 100 seconds free simulations of the fitted models ($10.3 \pm 2.7$ for the ground truth SNN) |
|---|---|---|
| neuLVM | $0.81 \pm 0.02$ | $7.8 \pm 4.1$ |
| PLDS | $(0.68 \pm 0.11)$ | not visible |
| SLDS | $(0.73 \pm 0.02)$ | $11.9 \pm 8.9$ |
| GLM | - | not visible |

Mean and ($\pm$) standard deviation were computed over 20 different trials. Parentheses for PLDS and SLDS indicate that these results are for the best linear transformation of the inferred latent variables.

estimated by simply summing and smoothing the observed spike trains (Appendix D) is shown in Figure S6.

On test (ii), neuLVM, fitted on a single trial of 10 seconds, was able to reproduce stochastic switches similar to that of the ground truth SNN (Table 1 second column): free simulations of the fitted neuLVM showed 7.8 switches in 100 seconds on average (10.3 switches on average for the ground truth SNN). To make sure that stochastic switches were the result of parameter learning via the Baum-Viterbi algorithm, we verified that, before learning, neuLVM did not show any metastable dynamics (Figure S7). Examples of simulated trials are shown in Figure 4B. PLDS failed to reproduce stochastic switches, which is not surprising since winner-take-all dynamics are typically nonlinear. SLDS could reproduce stochastic switches at the correct mean frequency (11.9 instead of the ground truth 10.3), but the standard deviation of the simulated switch count, 8.9 (2.7 for the ground truth SNN), indicates that a single 10 seconds trial was probably not sufficient for SLDS to learn switching probabilities reliably. Finally, neuronal stochasticity and small network size (9 neurons) did not allow GLM to produce stochastic switches, even when the training trial was prolonged to 500 seconds.

Taken together, only neuLVM could infer the latent population activity and reliably learn the metastable dynamics on single trials of 10 seconds, demonstrating the effectiveness of its neuronally-grounded inductive biases. Of course, these results do not guarantee that the inductive biases of neuLVM would be effective on real data since real data is most certainly out-of-distribution for neuLVM. While applications on real data are beyond the scope of this paper, in Appendix F, we show that neuLVM is robust, to a certain extent, to within-population heterogeneity and out-of-distribution data.

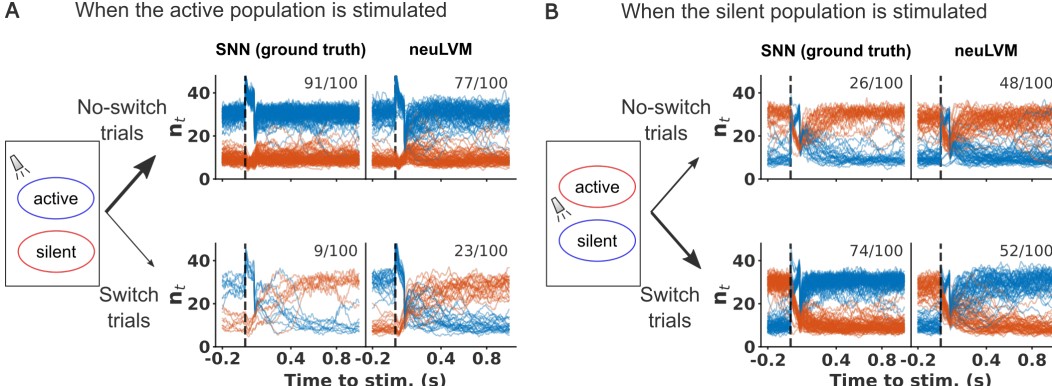

Figure 5: **Network responses to perturbations mimicking optogenetic stimulation.** **(A)** Activities of the excitatory populations when the active population is stimulated (100 trials, ratios indicate the number of No-switch or Switch trials). **(B)** Same as A but the silent population is stimulated.

### 5.2.2  Generalization: towards experimental predictions with neuronally-grounded modeling

Bottom-up, mechanistic models allow us to perform *in silico* experiments and generate predictions about neural microcircuits, which can then be tested experimentally. So we wondered: can neuLVM, fitted on a single trial of spontaneous activity (like in Section 5.2.1), predict the response of the SNN when an external perturbation is applied? As a preliminary step in that direction, we tested whether an external stimulation of the fitted model would generate the same response as that of the microscopic SNN when subjected to the same perturbation.

Using the same multi-population network as in Section 5.2 (Figure 3A) and neuLVM fitted on a single trial of spontaneous activity (Figure 3B), we compared the response of the ground truth SNN with that of neuLVM when one of the populations was stimulated by a current pulse of 4 ms mimicking the stimulation of an optogenetically modified population by a short light pulse. We simulated 100 trials where the momentarily active excitatory population was stimulated, and 100 where the momentarily silent excitatory population was stimulated (Figure 5A and B respectively). Each stimulation led to two possible outcomes: stimulation could trigger a state switch (Switch trials) or no state switch (No-switch trials). In both the ground truth SNN and the fitted neuLVM, we found that stimulating the silent population triggered more frequent state switches (Figure 5B) than stimulating the active population (Figure 5A). Moreover, in both the ground truth and the fitted neuLVM, we could induce 'excitatory rebound' switches by stimulating the active population (Figure 5A, lower half).

## 6   Discussion

Understanding the neural dynamics underlying computation in the brain is one of the main goals of latent variable modeling of multi-neuronal recordings [45, 47, 51, 53, 87, 88]. We contribute to this effort by proposing here a bottom-up, mechanistic LVM – the neuronally-grounded latent variable model (neuLVM) – which can be mapped to a multi-population SNN. Using SNN-based generative models, which are more biological than RNN-based models [45], could allow systems neuroscientists to test hypotheses about the architecture of the probed microcircuit, and provide a neuronally-grounded understanding of computation.

While this work shows the potential of the neuLVM approach, the application of neuLVM to real data faces two methodological challenges. First, there is the problem of identifiability: although neuLVM could recover a single unknown connectivity parameter (Section 5.1), our method could not always recover the SNN parameters when many parameters were unknown (Section 5.2). Bayesian inference could circumvent the problem of non-identifiability by estimating the full posterior distribution over model parameters [54, 89]. In addition, perturbing the probed network, with optogenetic stimulation, for example, could help model parameter recovery by providing richer data. Second, in the case of real data, choosing a good generative SNN model is a nontrivial task. For example, how many homogeneous populations should the SNN have? Clustering the recorded spike trains could guide the design of possible generative models and Bayesian model comparison, as used in biophysical

modeling of neuroimaging data [90–92], could help in selecting the most likely model among several possible models.

The model proposed here is only one particular example of an SNN-based, tractable latent variable model. Whether other such neuronally-grounded models of partially observed spike trains can be formulated and efficiently applied to real data is a question left for future work.

## Acknowledgments and Disclosure of Funding

We thank Johanni Brea for several discussions and for his comments on an early version of this work. We also thank Tilo Schwalger for the discussions and for sharing his code. Code from Joachim Koerfer was also used. This research was supported by Swiss National Science Foundation (no. 200020_184615).

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
