# Appendices of:

# Mesoscopic modeling of hidden spiking neurons

## A  Mesoscopic model in the case of LIF neurons

In this section, we present in detail the mesoscopic model of Schwalger et al. [11] in the case of multiple interacting populations of LIF neurons, as formulated in [13].

**Fine-grained SNN of LIF neurons with escape noise.**  Let us consider a general network of $N$ LIF neurons (indexed by $i = 1, \ldots, N$) with escape noise [10]. Neurons are modeled as point processes: the probability for neuron $i$ to emit a spike at time $t$, given the past network activity $\mathbf{y}_{1:t-1}$, is

$$p(y_t^i = 1 | \mathbf{y}_{1:t-1}, \Theta) = 1 - \exp\left(-\lambda_t^i \Delta t\right), \quad \text{with } \lambda_t^i = \exp\left(V^i(t|\hat{t}^i) - \vartheta^i\right),$$

where the escape rate (or stochastic intensity) $\lambda_t^i$ depends on the momentary difference between the membrane potential $V^i(t|\hat{t}^i)$ and the firing threshold $\vartheta^i$, via an exponential escape function. The voltage $V^i(t|\hat{t}^i)$ of neuron $i$ at time $t$ depends on its last spike time $\hat{t}^i = t - a^i$ and the inputs received up to time $t$, which include the inputs coming from the other neurons and the external input $\mathbf{I}_{1:t}^{\text{ext},i}$. Between spikes, for all $t > \hat{t}^i + t_{\text{ref}}^i$ ($t_{\text{ref}}^i$ being the absolute refractory period of neuron $i$), the voltage dynamics follows

$$V^i(t|\hat{t}^i) = V^i(t-1|\hat{t}^i) + \left(\frac{U_{\text{r}}^i + R I_t^{\text{ext},i} - V^i(t-1|\hat{t}^i)}{\tau_{\text{mem}}^i}\right) \Delta t + \sum_{j=1}^{N} J^{ij} \left(\epsilon^{ij} * \mathbf{y}^j\right)(t),$$

and $V^i(t|\hat{t}^i) = 0$, for all $t \le \hat{t}^i + t_{\text{ref}}^i$ (which means that the voltage is reset to 0 after each spike and is clamped at 0 for an absolute refractory period $t_{\text{ref}}^i \ge 0$). The parameters $\tau_{\text{mem}}^i > 0$ and $U_{\text{r}}^i > 0$ are the membrane time constant and the resting potential respectively. The neuron $i$ is therefore characterized by the parameters $\theta^i = \{\vartheta^i, U_{\text{r}}^i, \tau_{\text{mem}}^i, t_{\text{ref}}^i\}$. While the escape function is usually parameterized by a rescaled exponential function of the form $f(v) = \frac{1}{\tau_0^i} \exp(\beta^i(v - \tilde{\vartheta}^i))$ [10, Sec 9.1], the parameters $\tau_0^i, \beta^i$ and $\tilde{\vartheta}^i$ can be absorbed in $\vartheta^i$ (up to a rescaling of the resting potential $U_{\text{r}}^i$). The resistance $R = 1\,\Omega$ is used here simply for the consistency of physical units. The postsynaptic current induced by a spike of neuron $j$ on neuron $i$ is defined by the synaptic weight $J^{ij}$ and the synaptic kernel $\epsilon^{ij} : \mathbb{R}_+ \to \mathbb{R}_+$. In this work, we consider exponential kernels of the form $\epsilon^{ij}(t) = \frac{\mathcal{H}(t - \Delta^{ij})}{\tau_{\text{syn}}^{ij}} \exp\left(-\frac{t - \Delta^{ij}}{\tau_{\text{syn}}^{ij}}\right)$, where $\tau_{\text{syn}}^{ij}$ is the synaptic time constant, $\Delta^{ij}$ is the synaptic delay and $\mathcal{H}$ is the Heaviside function. The symbol $*$ denotes the convolution operator.

**Coarse-grained multi-population SNN.**  Coarse-graining and mean-field approximations consist in partitioning the $N$ neurons into $K$ homogeneous populations, indexed by $\alpha = 1, \ldots, K$, where (i) all the neurons $i$ in population $\alpha$ share the same neuronal parameters $\theta^i = \theta^\alpha$; (ii) for any neuron $j$ in population $\beta$ and any neuron $i$ in population $\alpha$, $J^{ij} = J^{\alpha\beta}/N^\beta$ ($N^\beta$ being the number of neurons in population $\beta$) and $\epsilon^{ij} = \epsilon^{\alpha\beta}$; (iii) all the neurons $i$ in population $\alpha$ share the same external input $\mathbf{I}^{\text{ext},i} = \mathbf{I}^{\text{ext},\alpha}$. In such a coarse-grained $K$-population SNN, we have, for any neuron $i$ in population $\alpha$,

$$\sum_{j=1}^{N} J^{ij} \left(\epsilon^{ij} * \mathbf{y}^j\right)(t) = \sum_{\beta=1}^{K} J^{\alpha\beta} \left(\epsilon^{\alpha\beta} * \mathbf{n}^\beta\right)(t)/N^\beta,$$

where $n_t^\beta = \sum_{i \in \text{pop. } \beta} y_t^i$ is the total number of spikes in population $\beta$ at time $t$. Hence, the probability for *any* neuron $i$ in population $\alpha$ to emit a spike at time $t$, given its age $a$ and the past

population activity $\mathbf{n}_{1:t-1}$ is

$$p_{t,a}^{\text{fire},\alpha} = 1 - \exp\left(-\lambda_t^\alpha \Delta t\right), \quad \text{with } \lambda_t^\alpha = \exp\left(V^\alpha(t|t-a) - \vartheta^\alpha\right). \tag{5}$$

For all $a > t_{\text{ref}}^\alpha$, we have the update rule

$$V^\alpha(t|t-a) = V^\alpha(t-1|t-a) + \left(\frac{U_{\text{r}}^\alpha + RI_t^{\text{ext},\alpha} - V^\alpha(t-1|t-a)}{\tau_{\text{mem}}^\alpha}\right)\Delta t$$
$$+ \sum_{\beta=1}^K J^{\alpha\beta}\left(\epsilon^{\alpha\beta} * \mathbf{n}^\beta\right)(t)/N^\beta,$$

and $V^\alpha(t|t-a) = 0$ for all $a \leq t_{\text{ref}}^\alpha$. This gives the explicit expression for the probability $p_{t,a}^{\text{fire}}$ in Eq. (1). In this work, for simplicity, we will assume that all the synaptic kernels are the same, i.e. $\epsilon^{\alpha\beta} = \epsilon, \forall \alpha, \beta$ (see Table S2).

**Mesoscopic description.** The $K$-population SNN described above does not by itself constitute a mesoscopic model because the probability $p_{t,a}^{\text{fire},\alpha}$ still involves the age $a$ of some neuron. To get a mesoscopic model (i.e. a model that does not involve the fine-grained modeling of each individual neuron), Schwalger et al. [11] used the population activity $\mathbf{n}$ to approximate the age density of each population and derived a closed-form system of stochastic integral equations: For all $\alpha \in 1, \ldots, K$,

$$n_t^\alpha \sim \text{Binomial}\left(N^\alpha, \ \bar{n}_t^\alpha/N^\alpha\right), \tag{6a}$$

$$\bar{n}_t^\alpha = \left[\sum_{a \geq 1} p_{t,a}^{\text{fire},\alpha} S_{t,a}^\alpha n_{t-a}^\alpha + \Lambda_t^\alpha\left(N^\alpha - \sum_{a \geq 1} S_{t,a}^\alpha n_{t-a}^\alpha\right)\right]_+, \tag{6b}$$

$$\Lambda_t^\alpha = \frac{\sum_{a \geq 1} p_{t,a}^{\text{fire},\alpha}(1 - S_{t,a}^\alpha)S_{t,a}^\alpha n_{t-a}^\alpha}{\sum_{a \geq 1}(1 - S_{t,a}^\alpha)S_{t,a}^\alpha n_{t-a}^\alpha}, \tag{6c}$$

where $S_{t,a}^\alpha = \prod_{s=0}^{a-1}(1 - p_{t-a+s,s}^{\text{fire},\alpha})$ is the survival, i.e. the probability for a neuron in population $\alpha$ to stay silent between time $t - a$ and $t - 1$. A concise version of the derivation of the mesoscopic model (6) is presented in [13]. Note that Eq. (6) is not a one-dimensional stochastic dynamical system: the Markov embedding of the stochastic dynamics (6) is infinite-dimensional [13]. Indeed, Eq. (6) does not only describes the evolution of the population activity $n_t^\alpha$ but it also describes the evolution of the whole age (pseudo) density $\{S_{t,a}^\alpha n_{t-a}\}_{a \geq 1}$ in the population, also called the "refractory density" [12].

Formally, the 'initial condition' of Eq. (6) is defined by the population activity $\mathbf{n}_t$ for all $t \leq 0$ (denoted $\mathbf{n}_{t\leq 0}$). Several practical choices of initial conditions have been discussed in [11, 13, 54]. In this work, if not otherwise specified, $\mathbf{n}_{t\leq 0}$ is taken to be time-invariant, with stationary activities estimated from the observed data (see below).

The size of the discrete time steps $\Delta t$ does not need to be the same for the fine-grained SNN and for the mesoscopic model (6). Indeed, it can be useful to take longer time steps for the mesoscopic description (time coarse-graining). In the following appendices, when there is an ambiguity, $\Delta t_{\text{meso}}$ will denote the time step length for the mesoscopic model and neuLVM. The length $\Delta t_{\text{meso}}$ will always be smaller or equal to the neuronal absolute refractory periods so that a neuron can fire at most once in each time step.

## B   neuLVM for multiple interacting populations

Let us assume that we observe, during $T$ time steps, the spike trains of $q$ simultaneously recorded neurons that are part of a $K$-population SNN of $N$ neurons, with $N > q$. For each of the population $\alpha = 1, \ldots, K$, $q^\alpha > 0$ neurons are observed ($\sum_{\alpha=1}^K q^\alpha = q$) and share the same set of neuronal parameters $\theta^\alpha$, input weights $\{J^{\alpha\beta}/N^\beta\}_{\beta=1}^K$, and output weights $\{J^{\beta\alpha}/N^\alpha\}_{\beta=1}^K$, where $N^1, \ldots, N^K$ are the numbers of neurons in each population ($\sum_{\alpha=1}^K N^\alpha = N$).

**The likelihood $\mathcal{L}$ of the observed spike trains.** Following the assumptions described above, the likelihood $\mathcal{L}$ of the observed spike trains $\mathbf{y^o}$ (a binary $q \times T$ matrix) can be formally written as $\sum_{\mathbf{n}} p(\mathbf{y^o}, \mathbf{n}|\Theta)$, where $\mathbf{n}$ (an integer-valued $K \times T$ matrix) is the population activity and $\Theta = \{\{J^{\alpha\beta}\}_{1 \leq \alpha, \beta \leq K}, \{\theta^\alpha\}_{\alpha=1}^K\}$ are the parameters of the $K$-population SNN. The probability $p(\mathbf{y^o}, \mathbf{n}|\Theta)$ factorizes in $T$ terms of the form

$$p(\mathbf{y}_t^\mathbf{o}, \mathbf{n}_t|\mathbf{y}_{1:t-1}^\mathbf{o}, \mathbf{n}_{1:t-1}, \Theta) = \underbrace{p(\mathbf{y}_t^\mathbf{o}|\mathbf{y}_{1:t-1}^\mathbf{o}, \mathbf{n}_{1:t-1}, \Theta)}_{\textbf{part a}} \underbrace{p(\mathbf{n}_t|\mathbf{n}_{1:t-1}, \Theta)}_{\textbf{part b}}.$$

The probability (**part a**) of the observed spikes $\mathbf{y}_t^\mathbf{o}$ at time $t$ given the past observed spike activity $\mathbf{y}_{1:t-1}^\mathbf{o}$ and the past population activity $\mathbf{n}_{1:t-1}$ is

$$p(\mathbf{y}_t^\mathbf{o}|\mathbf{y}_{1:t-1}^\mathbf{o}, \mathbf{n}_{1:t-1}, \Theta) = \prod_{\alpha=1}^K \prod_{i=1}^{q^\alpha} p(y_t^{\mathbf{o}, \alpha, i}|a^i, \mathbf{n}_{1:t-1}, \Theta) = \prod_{\alpha=1}^K \prod_{i=1}^{q^\alpha} p_{t, a^i}^{\text{fire}, \alpha},$$

where $p_{t, a^i}^{\text{fire}, \alpha}$, given by Eq. (5) in Appendix A, is the probability for the recorded neuron $i$ of population $\alpha$ to emit a spike at time $t$.

The probability (**part b**) of the population activity $\mathbf{n}_t$ at time $t$ given the past population activity $\mathbf{n}_{1:t-1}$ is

$$p(\mathbf{n}_t|\mathbf{n}_{1:t-1}, \Theta) = \prod_{\alpha=1}^K p(n_t^\alpha|\mathbf{n}_{1:t-1}, \Theta),$$

where $p(n_t^\alpha|\mathbf{n}_{1:t-1}, \Theta)$ is approximated by the mesoscopic model (6).

## C  Fitting algorithm for neuLVM

**Baum-Viterbi algorithm.** Given the observed spike trains $\mathbf{y^o}$, we optimize the likelihood $\mathcal{L} = \sum_{\mathbf{n}} p(\mathbf{y^o}, \mathbf{n}|\Theta)$ via an EM-like algorithm – the Baum-Viterbi algorithm [79]. Relying on the heuristic that the posterior $p(\mathbf{n}|\mathbf{y^o}, \Theta)$ should be concentrated around its maximum, we approximate the posterior $p(\mathbf{n}|\mathbf{y^o}, \Theta)$ by a point mass $\delta_\mu$, where $\mu = \arg\max_\mathbf{n} \log p(\mathbf{y^o}, \mathbf{n}|\Theta)$. By doing so, the alternating estimation (E) and maximization (M) step of the $n$-th iteration read

**E-step**. $\widehat{\mathbf{n}}^n = \text{argmax}_\mathbf{n} \log p(\mathbf{y^o}, \mathbf{n}|\widehat{\Theta}^{n-1})$,

**M-step**. $\widehat{\Theta}^n = \text{argmax}_\Theta \log p(\mathbf{y^o}, \widehat{\mathbf{n}}^n|\Theta)$.

**Details of the optimization.** In the **M-step**, parameters $\Theta$ are optimized using the L-BFGS-B algorithm and the optimization stops when either the maximum number of iterations ($\text{maxiter}_\text{M}$) is reached, or the objective function improves by less than $\text{ftol}_\text{M}$, or the maximum norm of the gradient is less than $\text{gtol}_\text{M}$. Hyper-parameters including $\text{maxiter}_\text{M}$, $\text{ftol}_\text{M}$ and $\text{gtol}_\text{M}$ are given in Table S5. In the **E-step**, to carry out gradient ascent, we approximate the discrete Binomial distribution Eq. (6a) by a Gaussian, i.e. $n_t^\alpha \sim \mathcal{N}(\bar{n}_t^\alpha, \bar{n}_t^\alpha)$, where $\bar{n}_t^\alpha$ is given by the mesoscopic model Eq (6) [11]. With this approximation, the latent population activity $\mathbf{n}$ is optimized with the Adam algorithm with learning rate $\text{lr}_\text{E}$ and the optimization stops when either the maximum number of iterations ($\text{maxiter}_\text{E}$) is reached, or the objective function stops improving for the last $\text{itertol}_\text{E}$ iterations. Hyper-parameters including $\text{lr}_\text{E}$, $\text{maxiter}_\text{E}$ and $\text{itertol}_\text{E}$ are given in Table S5. The estimated parameters $\widehat{\Theta}$ and the estimated latent population activity $\widehat{\mathbf{n}}$ are the result of many iterations of **E-step** and **M-step**. The fitting algorithm ends either when it stops improving the objective function or the maximum number of E-M iterations is reached.

**Multiple data-driven initializations.** To deal with the fact that the joint probability $p(\mathbf{y^o}, \mathbf{n}|\Theta)$ to optimize is non-convex and high-dimensional ($\mathbf{n}$ has dimension $K \times T$), we perform the Baum-Viterbi algorithm $\text{N}_\text{init}$ times with initial parameters $\widehat{\Theta}^0$ uniformly sampled in a certain range given in Appendices E and F. Since the sum over the observed neurons from population $\alpha$, $\sum_{i=1}^{q^\alpha} \mathbf{y}_{1:T}^{\mathbf{o}, i}$, already provides a rough estimate of the latent population activity $\mathbf{n}_{1:T}^\alpha$, the **E-Step** of the first iteration ($\widehat{\mathbf{n}}^1$) is replaced by an empirical estimation of the population activity $\widehat{\mathbf{n}}_\sigma^\text{sm}$ from the observed spike trains (see Appendix D).

**Numerical implementation of the mesoscopic model.** To implement the mesoscopic model (6), we approximate the infinite sums $\sum_{a \geq 1}$ in Eq. (6) by finite sums $\sum_{a=1}^{a_{\max}}$, where $a_{\max}$ is chosen to be large enough such that the probability for a neuron to remain silent for a duration longer than $a_{\max}$ is negligible. In our numerical implementation, the mesoscopic model (6) has therefore a finite memory $a_{\max}$. Note that a more principled way to implement finite memory can be found in [13], where a numerical implementation similar to ours is presented in detail. The hyper-parameter $a_{\max}$ is given in Appendices E and F. If not otherwise specified, the initial condition $\mathbf{n}_{\leq 0}$ of Eq. (6) are chosen to be time-invariant, with stationary activities estimated from the first $a_{\max}$ time steps of the recorded spike trains.

## D  Smoothed empirical population activity

A smoothed empirical estimation of the population activity $\widehat{\mathbf{n}}_\sigma^{\mathrm{sm}}$ was obtained from the recorded spike trains $\mathbf{y}^{\mathbf{o}}$ by applying a Gaussian smoothing kernel $g_\sigma$ with standard deviation $\sigma$. For population $\alpha = 1, \ldots, K$,

$$\widehat{\mathbf{n}}_{\sigma,t}^{\mathrm{sm},\alpha} = \left( \left( \frac{N^\alpha}{q^\alpha} \sum_{i=1}^{q^\alpha} \mathbf{y}_{1:T}^{\mathbf{o},\alpha,i} \right) * g_\sigma \right)(t).$$

## E  Details of the cluster state example

Values of parameters used in this example are given in Table S2, except if mentioned otherwise.

**When the network is initialized in the unstable asynchronous state (Figure 2B,C).** In this case, the network is always initialized, at time 0, in the same unstable asynchronous state with a firing rate of 20 Hz. The spike train power spectra (Figure 2B), for different choices of connectivity parameter $J$, were computed using 600 non-overlapping segments of 120 s. To measure the goodness of the connectivity recovered by newLVM, for each $J$ in $\{59, 60, 61, 62, 63, 64, 65\}$ mV, we simulated the ground truth SNN (starting from the same unstable asynchronous state mentioned above) for 1 s and further generated 10 different datasets with different samples of six observed neurons (1% of the population).

**When the network is initialized in a 4-cluster state (Figure 2D-F).** In this case, we simulated a trial (one second, $J = 60.32$ mV) with a transition from a metastable 4-cluster state to a 3-cluster state (Figure 2D,E). To test how well newLVM work in the regime where only a tiny fraction of the total number of neurons is observed, for each number $\{1, 2, 5, 10\}$ of observed neurons, we generated 10 different datasets with different samples of observed neurons.

**Fitting of the neuLVM.** The initial parameter $\widehat{J}^0$ was drawn uniformly in $[10, 30] \cup [90, 110]$ mV. The latent population activity was initialized as the smoothed empirical population activity ($\widehat{\mathbf{n}}^1 = \widehat{\mathbf{n}}_{\sigma,t}^{\mathrm{sm}}$, Appendix D) with $\sigma = 1.4$ ms (Figure 2E). Since the Baum-Viterbi algorithm converged reliably when only $J$ was unknown, $N_{\mathrm{init}}$ was set to 1. The hyper-parameter $\Delta t_{\mathrm{meso}}$ was set to 1 ms and $a_{\max}$ was set to 100 ($a_{\max} \Delta t_{\mathrm{meso}} = 100$ ms).

**Fitting of René et al. (2020).** A naive application of René et al. [54] consists in fitting the model with $\widehat{J} = \mathrm{argmax}_\Theta \log p(\widehat{\mathbf{n}}^1 | J)$. The parameter $\widehat{J}^0$ and the latent population activity $\widehat{\mathbf{n}}^1$ were set the same way as for neuLVM, but $N_{\mathrm{init}}$ was set to 200. The best performing $\widehat{J}$s were reported in Figure 2F. The hyper-parameters $\Delta t_{\mathrm{meso}}$ and $a_{\max}$ were the same as for neuLVM.

## F  Details of the metastable point attractors example

Values of parameters used in this example are given in Table S2, except if mentioned otherwise. In this example, we simulated a 500 s-long trial and randomly cut out 20 non-overlapping 10 s-segments to generate the training datasets.

**Fitting of the neuLVM.** The initial parameters $\widehat{\Theta}^0$ (which include the connectivities $J^{\cdot}$, membrane time constants $\tau_{\mathrm{mem}}^{\cdot}$, firing thresholds $\vartheta^{\cdot}$ and resting potentials $U_{\mathrm{r}}^{\cdot}$) were sampled randomly by assuming the uniform prior on the range $0.4$ to $2$ times the ground truth values. In this example, the connectivity matrix $\mathbf{J}$ was parametrized by $\{J^{\mathrm{e}1}\, J^{\mathrm{e}2}, J^{\mathrm{i}}\}_{J>0}$: $\mathbf{J} = \begin{pmatrix} J^{\mathrm{e}1} & 0 & 0 \\ 0 & J^{\mathrm{e}2} & 0 \\ 0 & 0 & J^{\mathrm{i}} \end{pmatrix} \begin{pmatrix} 1 & 0 & 1 \\ 0 & 1 & 1 \\ -1 & -1 & -1 \end{pmatrix}$ (see Figure 3A for the network architecture). The latent population activity was initialized as the smoothed empirical population activity ($\widehat{\mathbf{n}}^1 = \widehat{\mathbf{n}}_{\sigma,t}^{\mathrm{sm},\alpha}$, Appendix D) with $\sigma = 400$ms (Figure S7). Out of 5 fits ($\mathrm{N}_{\mathrm{init}} = 5$), the fit with the highest joint likelihood $p(\mathbf{y}^{\mathbf{o}}, \widehat{\mathbf{n}} | \widehat{\Theta})$ was selected. The related hyper-parameter $\Delta t_{\mathrm{meso}}$ was set to $4$ ms and $a_{\mathrm{max}}$ was set to $250$ ($a_{\mathrm{max}} \Delta t_{\mathrm{meso}} = 1000$ ms). When $\Delta t_{\mathrm{meso}}$ was set to a value that was larger than the $\Delta t$ of the recorded data, the recorded spike trains were downsampled.

**PLDS** We used code from `https://bitbucket.org/mackelab/pop_spike_dyn/src/master/`. To fit Poisson Linear Dynamical Systems (PLDS) [33] to the three-population example, we initialized the parameters with nuclear norm penalized rate estimation [8] and used the variational EM algorithm of [33]. The dimensionality of the latent states was set to three (the number of populations). The time resolution of the recorded spike trains was downsampled to $4$ ms ($\Delta t_{\mathrm{PLDS}} = 4$ ms). Other hyper-parameters were set to default.

**SLDS** We used code from `https://github.com/lindermanlab/ssm` [39]. To fit Poisson Switching Linear Dynamical Systems (SLDS) [34–38] to the three-population example, we updated the parameters with stochastic variational inference with the posterior approximated by a factorized distribution. The dimensionality of the continuous latent states was chosen to be three (the number of populations) and the dimensionality of the discrete latent states was chosen to be three (corresponding to the number of metastable states plus one for the transition state). We specified the 'emissions model' as 'Poisson_orthog' with the exponential escape function. Other hyper-parameters were set to default. Further, for SLDS to work, the discrete time step had to be large enough. Here we downsampled datasets to $40$ ms (the smallest $\Delta t_{\mathrm{SLDS}}$ that worked).

**An additional test.** We were interested to find out whether neuLVM is robust to within-population heterogeneity and slightly out-of-distribution data. To answer this question, we performed an additional test where we introduced within-population heterogeneity in the ground truth winner-take-all (WTA) network (Section 5.2) by adding noise to the connectivity and neuronal parameters as specified in the Table S6 (noise in the neuronal parameters is small to conserve metastable WTA dynamics). Furthermore, we set the $N$'s of the neuLVM to 300, 300, 300 (the $N$'s of the ground truth network are 400, 400, 200). We tested neuLVM on eight 10 s-segments cut out from a 100 s-long trial. The method is only mildly affected by these changes: all fitted neuLVM reproduced metastable WTA dynamics and the Pearson correlation between $\widehat{\mathbf{n}} | \mathbf{y}^{\mathbf{o}}$ and $\mathbf{n}^*$ was $0.76 \pm 0.02$, which is still higher than the correlations obtained by PLDS and SLDS (see Table 1).

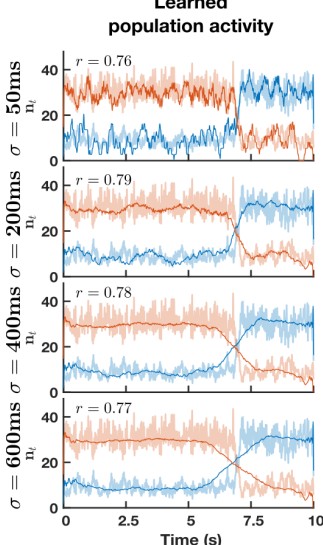

Figure S6: Smoothed empirical estimate $\hat{\mathbf{n}}_{\sigma,t}^{\mathrm{sm},\alpha}$ (Appendix D) of the latent population activity for one example trial (the same as in Figure 3, two excitatory populations). The value $r$ is the Pearson correlation coefficient between the inferred $\hat{\mathbf{n}}|\mathbf{y}^{\mathbf{o}}$ and the ground truth $\mathbf{n}^*$ population activities.

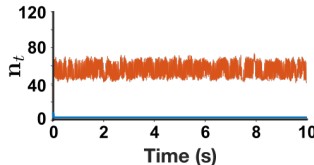

Figure S7: Spontaneous population activity simulated by the neuLVM before learning. Population activity of one excitatory population (the blue trace) quickly dies out. No visible metastable dynamics.

Table S2: Values of parameters used in simulations. **Boldface** is used to indicate fitted parameters.

| | Name | Description | Value | |
|---|---|---|---|---|
| | | | Example Section 5.1 Single excit. population | Example Section 5.2 Excitat. (inhib.) populations |
| | $\Delta t$ | time step | 1 ms | 0.2 ms |
| | $N$ | number of neurons | 600 | 400 (200) |
| $\Theta$ | $J$ | **connectivity** | **60.32 mV** | **9.984 mV (-19.968 mV)*** |
| | $\vartheta$ | **firing threshold** | **49.7 mV** | **3.7 mV (3.7 mV)** |
| | $U_{\mathrm{r}}$ | **resting potential** | **26 mV** | **14.4 mV (14.4 mV)** |
| | $\tau_{\mathrm{mem}}$ | **membrane time constant** | **100 ms** | **20 ms (20 ms)** |
| | $t_{\mathrm{ref}}$ | absolute refractory period | 0 ms | 4 ms (4 ms) |
| $\epsilon$ | $\tau_{\mathrm{syn}}$ | synaptic time constant | 4 ms | 3 ms (6 ms) |
| | $\Delta$ | synaptic delay | 10 ms | 0 ms (0 ms) |

\* i.e. for all population $\alpha$, $J^{\alpha\beta} = 9.984$ mV if $\beta$ is an excitatory population and $J^{\alpha\beta} = -19.968$ mV if $\beta$ is the inhibitory population.

Table S3: Performance summary (ii) when fitting neuLVMs to the single-population example (Section 5.2, Figure 2C) with $m$-cluster states. For each ground truth $J$, 10 different datasets were generated and tested. (6 observed neurons.)

| $J$ | 59 | 60 | 61 | 62 | 63 | 64 | 65 |
|---|---|---|---|---|---|---|---|
| $\widehat{J}$ (mean) | 58.18 | 60.46 | 61.20 | 62.05 | 62.39 | 63.58 | 63.66 |
| $\widehat{J}$ (std) | 0.50 | 0.71 | 0.93 | 0.46 | 1.11 | 1.59 | 1.85 |
| Pearson $r$ | | | | 0.81 ($p = 2.8\mathrm{e}{-17}$) | | | |

Table S4: Performance summary (i) when fitting neuLVMs to the single-population example (Section 5.2, Figure 2F) with a transition from a metastable 4-cluster state to a 3-cluster state. For each ground truth $J$, 10 different datasets were generated and tested. ($J = 60.32$ mV.)

| # observed neurons | 1 | 2 | 5 | 10 | 599 |
|---|---|---|---|---|---|
| $\widehat{J}$ (mean) | 60.37 | 60.14 | 59.33 | 59.81 | 59.10 |
| $\widehat{J}$ (std) | 2.43 | 1.48 | 0.91 | 1.25 | 1.26 |

Table S5: Hyper-parameters used when fitting neuLVM.

| Name | Value | |
|---|---|---|
| | Example Section 5.1 Single excit. population | Example Section 5.2 Excitat. (inhib.) populations |
| $\mathrm{lr_E}$ | 1e−3 | 1e−3 |
| $\mathrm{maxiter_E}$ | 200 | 200 |
| $\mathrm{itertol_E}$ | 3 | 3 |
| $\mathrm{lr_M}$ | 1e−8 * | 1e−8* |
| $\mathrm{maxiter_M}$ | 200 | 200 |
| $\mathrm{ftol_M}$ | 2e−9* | 2e−9* |
| $\mathrm{gtol_M}$ | 1e−5* | 1e−5* |

* Values are default as in scipy.optimize.minimize(method='L-BFGS-B').

Table S6: Within-population heterogeneity introduced in the ground truth winner-take-all (WTA) network (in the additional experiment of Appendix F).

| ground truth within-population heterogeneity | $\mu$ | $\sigma$ (normal distribution) |
|---|---|---|
| $J^{\mathrm{e}^1} J^{\mathrm{e}^2}, J^{\mathrm{i}}$ | 9.98 / 9.98 / 19.97 | 2.00 / 2.00 / 2.00 |
| $\vartheta$ | 3.70 | 0.07 |
| $U_{\mathrm{r}}$ | 14.40 | 0.29 |
| $\tau_{\mathrm{mem}}$ | 20.00 | 0.40 |