# OpenReview forum: "Mesoscopic modeling of hidden spiking neurons"
_NeurIPS.cc/2022/Conference — NeurIPS 2022 Accept_

### Official Review · Reviewer_Vyef · 2022-07-08

**Rating:** 8
**Confidence:** 4
**Soundness:** 4 excellent
**Presentation:** 4 excellent
**Contribution:** 3 good

**Summary:**

The paper proposes a latent variable model for modeling observed activity in spiking neural networks while taking into account the activity of unobserved neurons. To do this, the authors propose a mean-field approximation approach to reduce the effect of all unobserved neurons to a lower dimensional summary quantity with simplified parameters. The proposed model, neurLVM, is fit via the approximate “hard EM” algorithm. The model is validated in two simulations, where it provides accurate recovery of net population activity and of switching states. The fitted, reduced neurLVM model was also able to predict the effect of stimulation on the true simulated network.

**Questions:**

1. In the conclusion, it is stated that neurLVM cannot always recover the SNN parameters when many parameters are unknown. I appreciate the authors including this detail as a limitation. However, it was not clear to me what the learned parameters were for the metastable point attractors example. Could the authors clarify this and/or make sure those parameters are reported?

2. In the three-population example, it appears that the cluster identities of the nine observed neurons were obtained from clustering all of the neurons (that is, including the unobserved neurons). If that is the case, can you identify the correct clustering just from the 9 observed neurons?

3. As mentioned in the previous section, it is not clear how the method performs on heterogeneous populations. My understanding is that in both sets of simulations, the populations or subpopulations are homogeneous. It would be great to evaluate in a setting where the individual neuron parameters vary within a population, e.g. they are drawn from a distribution over the parameters.

4. I’m curious on what the authors think about the following question. How restrictive is it for the sum of spikes within a population to be the primary variable of interest? That is, the latent population activity is reduced to a one-dimensional variable. I understand that was important for model fitting. But for example, does that limit the method to identifying only one-dimensional summaries of populations?

5. Have the authors considered a different approximate EM algorithm that maintains an approximate distribution over the latent spiking activity? E.g., a variational EM approach?


**Limitations:**

The authors discussed two limitations, parameter recovery and choices of parameters and architecture for applications to real data, in their conclusion.

**Strengths And Weaknesses:**

The paper is clear and of high-quality. It provides an original contribution for including the effect of unobserved neurons when fitting spiking neural network models. This is a significant step forward in models of neural spiking responses. The proposed methods and experiments were clearly described with sufficient detail. Additionally, the model showed good performance in the simulations.

One weakness is the homogeneous population assumption seems to limit the generality of the method. It is not clear how the approach would work for heterogeneous populations with higher dimensional activity. Nonetheless, I think the proposed method is an important step forward.

=== Update ===

I appreciate the authors additional experiments and responses to my questions. In particular, I think including the heterogeneous population experiment strengthens the evaluation of the proposed approach. Based on the additional evaluations and clarity, I have increased my score by a point.

---

> ### Author Response · Authors · 2022-08-02
> **Point-by-point responses**
>
> We thank the reviewer for her/his assessment. We have addressed important and overlapping questions in the common post to all reviewers. In particular, the reviewer may be interested in
> * Q1: Is neuLVM robust to within-population heterogeneity and slightly out-of-distribution data?
> * Q2-3: Details about parameter inference for the metastable point attractors example.
>
> Here are our point-by-point responses to her/his questions.
>
> 1. *In the conclusion, it is stated that neurLVM cannot always recover the SNN parameters when many parameters are unknown. I appreciate the authors including this detail as a limitation. However, it was not clear to me what the learned parameters were for the metastable point attractors example. Could the authors clarify this and/or make sure those parameters are reported?*
>     * See Q2, Q3, and Q4.
>
> 2. *In the three-population example, it appears that the cluster identities of the nine observed neurons were obtained from clustering all of the neurons (that is, including the unobserved neurons). If that is the case, can you identify the correct clustering just from the 9 observed neurons?*
>     * In the paragraph the reviewer is referring to, we do a first clustering test with 25 observed neurons. Although we did not report it, we did an analogous test with 9 observed neurons, which was also successful. Note that this success is not surprising since that, in the considered example, the cluster identities can be seen with the naked eye already. If the reviewer finds this paragraph confusing, we could shorten it, as the clustering test is not essential. If the reviewer requests it, we will make this paragraph clearer to avoid any confusion.
>
> 3. *As mentioned in the previous section, it is not clear how the method performs on heterogeneous populations. My understanding is that in both sets of simulations, the populations or subpopulations are homogeneous. It would be great to evaluate in a setting where the individual neuron parameters vary within a population, e.g. they are drawn from a distribution over the parameters.*
>     * See Q1. In short, heterogeneity in connectivities and neuronal parameters could indeed compromise the performance of neuLVM. However, if heterogeneity is modest, neuLVM can still reproduce the dynamics of the ground truth. Therefore, if the populations are only mildly inhomogeneous, we expect a good performance from neuLVM. Note that i.i.d. noise in the synaptic weights is generally not too problematic because a neuron has many input weights and this noise tends to average out. A thorough investigation of the impact of inhomogeneity on the performance of neuLVM is beyond the scope of this work, but would be definitely interesting.
>
> 4. *I’m curious on what the authors think about the following question. How restrictive is it for the sum of spikes within a population to be the primary variable of interest? That is, the latent population activity is reduced to a one-dimensional variable. I understand that was important for model fitting. But for example, does that limit the method to identifying only one-dimensional summaries of populations?*
>     * This is a very good question, which, in our understanding, also questions the use of mean-field approximations in neuroscience. Regarding neuLVM, we should stress one important point. While it is correct to say that neuLVM infers the sum of spikes, we can nevertheless compute the distribution over the voltages in the population at a certain time from the *history* over the sum of spikes (see Appendix A). Therefore, neuLVM also infers the distribution over the voltages, which is formally an infinite-dimensional variable. The fact that neuLVM infers at the same time a one-dimensional variable and an "infinite-dimensional variable" reflects the nature of the mesoscopic model of (Schwalger et al. 2017): it is an infinite-dimensional Markov process with a one-dimensional value of interest. Our point is that neuLVM does not only identify one-dimensional summaries of populations; it actually identifies the state of the population, i.e. the distribution over the voltages (see Appendix A). We plan to illustrate this point in the appendix.
>
> 5. *Have the authors considered a different approximate EM algorithm that maintains an approximate distribution over the latent spiking activity? E.g., a variational EM approach?*
>     * The mesoscopic model, since it is non-Markovian and nonlinear, makes variational EM approaches nontrivial and computationally too expensive (unlike for PLDS). Therefore, after the analysis of the computational complexity of this approach, we did not implement it.
>
>
> We thank the reviewer for reading our responses. We hope the reviewer finds them useful. Also, please let us know if we haven’t fully addressed your concerns - we will be happy to address them further.

---

### Official Review · Reviewer_K3UJ · 2022-07-11

**Rating:** 7
**Confidence:** 4
**Soundness:** 3 good
**Presentation:** 4 excellent
**Contribution:** 4 excellent

**Summary:**

In the paper "Mesoscopic modeling of hidden spiking neurons" the authors introduce a neuronally-grounded latent variable model for fitting populations of observed and hidden units. The latter are described at the mesoscopic level. The model is evaluated on synthetic data with a single homogeneous population and with multiple populations and compared against other competing methods.

**Questions:**

The model assumes that the hidden populations are homogeneous with respect to both the recurrent synaptic weights and the single neuron parameters. I wonder how limiting this assumption is. How robust is the model to violations of this assumption? The synthetic data was generated with perfectly homogeneous populations. Would the recovered recurrent synaptic weight be close to the average ground truth parameters if there were some noise on the weights and single neuron parameters? How would neuLVM perform compared to PLDS and SLDS in the case of somewhat inhomogeneous hidden populations?

In the Introduction, the authors state that from the perspective of a single recorded neuron the spike activity of all the other neurons can be reduced to a single causal variable, further specified as the total recurrent input (line 27). This is true for typical single neurons in SNNs but is not true for real biological neurons or realistic multicompartment models, given all the complex intraneuron dynamics where spikes from particular other neurons can matter. The statement should be made more specific.

**Limitations:**

The authors did not discuss any societal impact. This is appropriate for this work.

The authors briefly discuss limitations regarding non-identifiability and potential need for preprocessing. However, given the seemingly strong assumption of homogeneous hidden populations, I would have expected a more thorough discussion of limitations.

**Strengths And Weaknesses:**

The authors address a very difficult problem, the problem of dealing with unobserved populations in spiking neural networks, which has plagued SNN modeling for decades. They propose a very promising approach, introducing inductive biases from biology to model hidden populations, but still keeping the level of description of these populations coarse. Therefore, significance of this work is excellent.

Clarity of the paper and figures is excellent too. The text is very well written and organized focusing on the single hidden population in the main paper and the more general case in the appendix. All equations are clear with symbols well explained.

The work builds on work on mesoscopic modeling of SNNs by Rene et al. and extends this to a latent variable model, which is trained by the Baum-Viterbi algorithm. This is an interesting and novel approach.

The evaluation of the model shows some of its strengths when compared to PLDS, SLDS and GLMs but falls short of convincing me that this would hold in more realistic cases where there is some variability in mostly homogeneous hidden populations. See Questions and Limitations below.

The authors do not see their method as being ready for application to real data yet (c.f. Discussion). Nevertheless, this work is an important stepping stone to deal with unobserved population activity.

The authors haven't made their source code available! To my understanding, the source code should be available to reviewers.

---

> ### Author Response · Authors · 2022-08-02
> **Point-by-point responses**
>
> We thank the reviewer for her/his assessment. We have addressed important and overlapping questions in the common post to all reviewers. In particular, the reviewer may be interested in
> * Q1: Is neuLVM robust to within-population heterogeneity and slightly out-of-distribution data?
> * Q5: How could neuLVM be applied on real data?
>
> Regarding the source code, we regret we did not make it available to the reviewers (which is recommended but not mandatory). We will follow this recommendation in the future. Of course, we will make the code of this work available, at the latest after publication.
>
> Here are our point-by-point responses to her/his questions.
>
> 1. *The model assumes that the hidden populations are homogeneous with respect to both the recurrent synaptic weights and the single neuron parameters. I wonder how limiting this assumption is. How robust is the model to violations of this assumption? The synthetic data was generated with perfectly homogeneous populations. Would the recovered recurrent synaptic weight be close to the average ground truth parameters if there were some noise on the weights and single neuron parameters? How would neuLVM perform compared to PLDS and SLDS in the case of somewhat inhomogeneous hidden populations?*
>     * See Q1. In short, heterogeneity in connectivities and neuronal parameters could indeed compromise the performance of neuLVM. However, if heterogeneity is modest, neuLVM can still reproduce the dynamics of the ground truth. Therefore, if the populations are only mildly inhomogeneous, we expect a good performance from neuLVM. Note that i.i.d. noise in the synaptic weights is generally not too problematic because a neuron has many input weights and this noise tends to average out (law of large numbers). A thorough investigation of the impact of inhomogeneity on the performance of neuLVM is beyond the scope of this work, but would be definitely interesting.
>
> 2. *In the Introduction, the authors state that from the perspective of a single recorded neuron the spike activity of all the other neurons can be reduced to a single causal variable, further specified as the total recurrent input (line 27). This is true for typical single neurons in SNNs but is not true for real biological neurons or realistic multicompartment models, given all the complex intraneuron dynamics where spikes from particular other neurons can matter. The statement should be made more specific.*
>     * We thank the reviewer for pointing this out. We will specify in the manuscript that this holds if neurons are seen as simplified point neurons.
>
> We thank the reviewer for reading our responses. We hope the reviewer finds them useful. Also, please let us know if we haven’t fully addressed your concerns - we will be happy to address them further.

---

> > ### Comment · Reviewer_K3UJ · 2022-08-04
> > **Recommending acceptance**
> >
> > I thank the authors for their detailed responses to my concerns. In particular the evaluation in response to the robustness question is very informative and clarifies the capabilities of the model. I am also happy to hear that the authors will release their source code once the paper is accepted.
> >
> > I do not have any additional questions about this work.
> >
> > Overall, I still feel that my rating is appropriate for this paper: I recommend acceptance since this is a solid paper..

---

### Official Review · Reviewer_k4DX · 2022-07-11

**Rating:** 7
**Confidence:** 4
**Soundness:** 3 good
**Presentation:** 3 good
**Contribution:** 3 good

**Summary:**

The authors propose a new type of latent space model for neural spike trains, based on a spiking neural network (SNN). They use mean field approximations to abstract parts of the SNN, resulting in the latent dynamics, but keep the initial formulation for the observed neurons. The authors show, that after pre-defining a single or multiple neural clusters they can recover the connectivity value(s) by an EM algorithm from snippets of 10 sec activity of only some observed neurons. The proposed latent model is able to reproduce some key functionality of the SNN and outperforms other latent variable benchmark models.

**Questions:**

1. It is surprising that the number of observed neurons seem to have (almost) no influence on the estimation performance (Fig. 2F / S4). Can the authors give an explanation for this?
2. A potential application of the method to experimental data poses several questions:
   - How is the number of clusters obtained?
   - How are the observed neurons assigned to the clusters?
   - What would be an appropriate ‘overall connectivity’ $A$ in $J = J^* \cdot A$ (Section E) for the assumed clusters?
3. Generalization (l. 285 ff):
   - (a) It is not clear to me how a “perfectly fitted” neuLVM would behave under the performed additional stimulations. Could the authors comment on this?
   - (b) The numbers in Fig. 5 suggest quantitatively quite different behaviours from the ground truth. Is this because of different parameter fits, or inherently due to the mesoscopic model?
4. Could a more general neural model simply be plugged into the neuLVM instead of a LIF? As proposed in the Discussion (l. 312). How would adaptational processes in the SNN / different interactions influence the proposed mesoscopic approach?

**Limitations:**

The authors mention two shortcomings before applying their method to experimental data. However there are more open questions and limitations which could be commented on (see also questions). The authors did not comment on potential negative societal impact of their work.


**Strengths And Weaknesses:**

### Strengths
The paper is clearly written and the authors define the posed problem nicely. The work is technical sound and the experiments are worked out thoroughly. The proposed latent model is an interesting approach to bridge the gap between pure statistical models and more biologically interpretable models. It has potentials for further investigations and the applications to experimental data.

However, there are several weaknesses:

### Weaknesses
1. The authors test their setup only on one parameter set, it is not clear that the results are robust to different network configurations.
2. The comparison which the authors make between the different models is quite unfair as the ground truth data is generated from the microscopic model the neuLVM is based on. They share therefor the same inductive biases whereas the other three models could be advantageous on differently generated data.
3. A demonstration on real experimental would have been nice. But this is potentially beyond the scope of this manuscript.
4. There are several open questions (see questions), especially when it comes to an application to experimental data.
5. The model makes quite some assumptions on the network structure (number of E/I clusters, base connectivity pattern) etc. This is a limitations which the authors should discuss in more detail.

### Minor comments:
- Fig. 2F: A log scale for the y-axis could be more appropriate.

### Update:
After considering the responses to this review and the reviews of the other reviewers the score was raised by +1.

---

> ### Author Response · Authors · 2022-08-02
> **Point-by-point responses**
>
> We thank the reviewer for her/his assessment. We have addressed important and overlapping questions in the common post to all reviewers. In particular, the reviewer may be interested in
> * Q4: In the WTA experiment (Section 5.2), can the parametrization of connectivities J’s be relaxed?
> * Q5: How could neuLVM be applied to real data?
>
> Here are our point-by-point responses.
>
> 1. (Weakness-1) *The authors test their setup only on one parameter set, it is not clear that the results are robust to different network configurations.*
>     * In the single-population experiment (Section 5.1), we systematically assess the ability of neuLVM to recover the connectivity parameter $J$, for seven different ground truth $J$ values (Figure 2C) corresponding to qualitatively different population dynamics (Figure 2B).
>
> 2. (Weakness-2) *The comparison which the authors make between the different models is quite unfair as the ground truth data is generated from the microscopic model the neuLVM is based on. They share therefor the same inductive biases whereas the other three models could be advantageous on differently generated data.*
>     * If the comparisons of neuLVM against the other models are seen from a purely "goodness-of-fit" perspective, we agree with the reviewer. The comparisons with the method of René et al., PLDS, and SLDS are mainly done to illustrate the conceptual difference between our approach and previous approaches. Importantly, to our knowledge, neuLVM has no direct competitor since its purpose is to provide a SNN-based mechanistic *interpretation* to partially observed spike trains, under some coarse-graining (homogeneous populations) assumptions (note that previous approaches based on simulations of the detailed SNN would have a prohibitive computational cost in our setups).
>
> 3. (Weakness-5) *The model makes quite some assumptions on the network structure (number of E/I clusters, base connectivity pattern) etc. This is a limitations which the authors should discuss in more detail.*
>     * See Q4.
>
> 4. (Question-1) *It is surprising that the number of observed neurons seem to have (almost) no influence on the estimation performance (Fig. 2F / S4). Can the authors give an explanation for this?*
>     * The single-population experiment, where only the connectivity parameter $J$ has to be inferred, is an easy task for neuLVM. Two observed neurons suffice for neuLVM to 'solve' this task, leaving little room for improvement when more neurons are observed. This illustrates the effectiveness of the inductive bias.
>
> 5. (Question-2) *A potential application of the method to experimental data poses several questions:
> How is the number of clusters obtained?
> How are the observed neurons assigned to the clusters?
> What would be an appropriate ‘overall connectivity’  in  (Section E) for the assumed clusters?*
>     * For the first two questions, see Q5. The overall connectivities could be constrained such as to obey Dale's law, or any other *a priori* biological constraint. This is just a suggestion.
>
> 6. (Question-3) *Generalization (l. 285 ff):
> (a) It is not clear to me how a “perfectly fitted” neuLVM would behave under the performed additional stimulations. Could the authors comment on this?
> (b) The numbers in Fig. 5 suggest quantitatively quite different behaviors from the ground truth. Is this because of different parameter fits, or inherently due to the mesoscopic model?*
>     * A "perfectly fitted" neuLVM would have the same (probabilistic) behavior as the ground truth SNN. The quantitative difference in probabilities are due to different parameter fits (See Q3). The mesoscopic model is very accurate in the case of LIF neurons.
>
> 7. (Question-4) *Could a more general neural model simply be plugged into the neuLVM instead of a LIF? As proposed in the Discussion (l. 312). How would adaptational processes in the SNN / different interactions influence the proposed mesoscopic approach?*
>     * While we derived neuLVM in the case of LIF neurons, neuLVM's can be derived for any neuron model considered in (Schwalger et al. 2017). This includes LIF with spike-frequency adaption (or Generalized LIF) and Generalized Linear Model/Spike Response Model neurons. For these later models (which can have adaptation), the mesoscopic approach relies on the "quasi-renewal approximation" of (Naud and Gerstner, 2012).
>
>
> We thank the reviewer for reading our responses. We hope the reviewer finds them useful. Also, please let us know if we haven’t fully addressed your concerns - we will be happy to address them further.

---

> > ### Comment · Reviewer_k4DX · 2022-08-05
> > **Response:**
> >
> > Thanks for the detailed answers and explanations. I have no further questions, but some loose comments:
> >
> > - There are still some questions on how to apply the method to experimental data, but for the current manuscript, it is sufficient in my opinion.
> > - Response to Q3 is a bit worrying, as the authors argue to have a (biologically) interpretable model. Non-identifiability counteracts the wish to draw biological solutions, but identification of the full posterior would indeed help.
> > - Response to Q4 is extremely interesting and leaves room for more investigation, for example, to find maximally informative perturbations.
> >
> > Overall, I am even more convinced that this is a very solid paper and should be accepted. Therefore I will raise my score by one.

---

### Official Review · Reviewer_1XR1 · 2022-07-11

**Rating:** 6
**Confidence:** 5
**Soundness:** 3 good
**Presentation:** 3 good
**Contribution:** 3 good

**Summary:**

New tool for inferring underlying properties of partially observed spiking neural networks (generalization of the fully observable solution by Rene et al), based on mean field modeling on net effect of unobserved neurons; validated on within model class simulated data.

**Questions:**


- fig 2F: why is the NeuLVM comparison restricted to only up to 10 observed neurons?
- confused about the logic of 'confirming clustring structure' and the making sure that the downsampling has the same amount of neurons from each subpopulation: please explain. This kind of selection bias does not seem to reflect real world data
- what does it mean that the 'neural paramaters were given in 5.1'? only J is treated as a parameter, everything else has ground truth values?
- i do not understand the quantification of state switched, in particular for PLDS and SLDS
- what does 'free simulations' mean? samples from the graphical model with data inferred parameters?
- is the poisson noise and/or model mismatch the cause for spike sample differences? i.e. the models (or at least PLDS assumes stationarity and is not designed to handle switches, so model mismatches are accounted for by higher overall noisiness?
- how fair is the across model comparison given that the ground truth data matched your model assumptions perfectly whereas it is out of model distribution for everything else?
- can you comment more about model degeneracies when fitting all model parameters rather than just within and across population Js

**Limitations:**

Technical limitations of the applicability of the procedure to real data should be discussed a lot more. No ethical issues.

**Strengths And Weaknesses:**

Strengths:
- clear setup and motivation
- novelty: interesting attempt at extracting more interpretable latent models, with a focus on modeling spiking activity of neurons (including unobserved ones)
- nice mix of traditional comp neuro and ML estimation

Weaknesses:
-  writing clarity: i would have appreciated a clear spelling out of the graphical model, separation between what counts as observations, inferred latent variable and model parameters (the current text mixes in mean field technicalities which makes it less clear than it should be)
in particular the link between n and y (Eq.4b back-referring to Eq.1) was difficult to get from the text, but i found both section 3 and 4 meandering and hard to follow in places
- the interpretability of the parameters may quickly become problematic for out of distribution data, in particular when it's not clear that the observed neural responses can be easily partitioned in a small set of homogeneous subpopulations (especially given the extremely strict notion of homogeneity required here); arguably heterogeneity if a key feature of brain circuits, yet is not clear how sensitive is the estimation to deviations from the strict homogeneity assumption
- metastable dynamics deviate by construction from the model assumptions of the most commonly used latent dynamical systems models, making them a perhaps unfair choice as the main benchmark for comparison across models.
- simulations are often somewhat anecdotal (fig 2DE)
- comparison to the fully observed scenario is rather trivial. if the model assumptions are radically different from the ground truth it is unreasonable to assume its estimated parameters to match the data

Minor:
-'photo-stimulation' is nonstandard terminology, especially if talking to an experimentalist -- causal manipulation, optogenetics etc would prove more useful

---

> ### Author Response · Authors · 2022-08-02
> **Point-by-point responses**
>
> We thank the reviewer for her/his assessment. We have addressed important and overlapping questions in the common post to all reviewers. Below are our point-by-point responses to her/his questions.
>
> 1. *fig 2F: why is the NeuLVM comparison restricted to only up to 10 observed neurons?*
>     * The neuLVM already performs well when the number of observed neurons is small: the inferred $\widehat{J}$'s already have an accuracy comparable with that obtained in the fully observed case (see Table S4). Observing a few neurons only was not due to computational cost.
>
> 2. *Confused about the logic of 'confirming clustering structure' [...]*
>     * In the paragraph the reviewer is referring to, our purpose is to verify that *if* the ground truth network has a multi-population architecture (with multiple homogeneous populations), clustering the observed neurons, using k-means clustering and the van-Rossum distance, is sufficient to identify the number of homogeneous populations.
>     * Downsampling to exactly $3$ observed neurons per population is an arbitrary choice we made to construct the artificial dataset; the method should work similarly well with different numbers of observed neurons per population. In the case where the number of neurons per population is uneven, re-weighted likelihood functions could be considered. If the reviewer requests it, we will make this paragraph clearer to avoid any confusion.
>
>
> 3. *what does it mean that the 'neural paramaters were given in 5.1'?[...]*
>     * Yes, exactly. We will improve the formulation in the text.
>
> 4. *i do not understand the quantification of state switched, in particular for PLDS and SLDS*
>     * For neuLVM, we count the number of times the inferred latent population activity of the excitatory populations changes from population 1 being active to population 2 being active and vice versa, during free simulations (i.e. samples of the graphical model). During free simulations of PLDS and SLDS, we do the same count but on the best linear transformation of the simulated latent variables onto the three population activities. The best linear transformation was estimated during training (test (i)).
>
>
> 5. *what does 'free simulations' mean? [...]*
>     * Yes, exactly. Free simulations are samples from neuLVM (the graphical model) with data inferred parameters. We will further clarify it in the main text if necessary.
>
> 6. *is the Poisson noise and/or model mismatch the cause for spike sample differences?[...]*
>     * For PLDS, we believe the model mismatch is indeed the main cause for spike sample differences, and the mismatch is accounted for by higher noisiness.
>     * SLDS, which is specifically designed to handle switches, still falls short of reaching the performance of neuLVM on test (i) (inference of the latent population activities) most likely because the data is slightly out-of-distribution.
>     * Regarding the Poisson noise, we are not sure whether the reviewer refers to the Poisson noise in the observations or in latent dynamics. If the former, all three models use Poisson noise for the observations. If the latter, even neuLVM has approximately Gaussian noise: the Poisson noise in the mesoscopic model (modeled as a binomial random variable in each time step, Eq (2a)) is actually approximately Gaussian due to the relatively large number of neurons (this is also explained in Schwalger et al, 2017).
>
> 7. *how fair is the across model comparison [...]*
>      * In Q1, we give an example where adding heterogeneity in the ground truth network and having a mismatch between the number of neurons per population $N$'s in neuLVM and in the ground truth -- both contributing to making the data out-of-distribution -- does not significantly affect the performance of neuLVM.
>     * If the comparisons of neuLVM against the other models are seen from a purely "goodness-of-fit" perspective, then they are indeed unfair, since the data is out-of-distribution for the other models. The comparisons with the method of René et al., PLDS and SLDS are mainly done to illustrate the conceptual difference between our approach and previous approaches. Importantly, to our knowledge, neuLVM has no direct competitor since its purpose is to provide a SNN-based mechanistic *interpretation* to partially observed spike trains, under some coarse-graining (homogeneous populations) assumptions (note that previous approaches based on simulations of the detailed SNN would have a prohibitive computational cost in our setups).
>
> 8. *can you comment more about model degeneracies [...]*
>     * See Q2, Q3, and Q4.
>
> We thank the reviewer for reading our responses. We hope the reviewer finds them useful. Also, please let us know if we haven’t fully addressed your concerns - we will be happy to address them further.

---

### Author Response · Authors · 2022-08-02
**Common responses to all reviewers**

We thank all reviewers for their thorough reviews. Following their comments, we performed additional tests and propose some changes to the text. Below, we respond to important questions asked by several reviewers.

#### Q1: Is neuLVM robust to within-population heterogeneity and slightly out-of-distribution data?
To answer this question, we performed an additional test, which will be added to the appendix. We introduced within-population heterogeneity in the ground truth winner-take-all (WTA) network (Section 5.2) by adding noise to the connectivity and neuronal parameters as specified in the following table (noise in the neuronal parameters is small to conserve metastable WTA dynamics).
| ground truth within-population heterogeneity | $\mu$ | $\sigma$ (normal distribution) |
| - | - | - |
| $J^{e^1}$  / $J^{e^2}$ / $J^{i}$ | 9.98 / 9.98 / 19.97 | 2.00 / 2.00 / 2.00 |
| $\vartheta$ | 3.70 | 0.07 |
| $U_{r}$  | 14.40  | 0.29  |
| $\tau_{mem}$  | 20.00  | 0.40 |

Furthermore, we set the $N$'s of the neuLVM to $\{300,300,300\}$ (the $N$'s' of the ground truth network are $\{400,400,200\}$). We tested neuLVM on eight 10 s-segments cut out from a 100 s-long trial. The method is only mildly affected by these changes: all fitted neuLVM reproduced metastable WTA dynamics and the Pearson correlation between $\widehat{\mathbf{n}}|\mathbf{y^o}$ and $\mathbf{n}^*$ was 0.76 $\pm$ 0.02, which is still higher than the correlations obtained by PLDS and SLDS (see Table 1 in the main text).

#### Q2: Are the within- and across-populations $J$'s the only parameters being inferred?

No, in the winner-take-all experiment (Section 5.2), we also infer the neuronal parameters (see Appendix A). We will make sure that the text is unambiguous on this point.

#### Q3: In the WTA experiment (Section 5.2), how well are ground truth parameters recovered?

The ability of neuLVM to recover the ground truth parameters largely depends on the identifiability of the model. In the WTA example, the model is probably not identifiable partly because recordings consist mostly of (metastable) stationary activity. To provide some insight on the parameter recovery ability of neuLVM in the WTA example, we performed dedicated tests, which will be reported in the appendix. In summary, the excitatory $J^{e^1}$ and $J^{e^2}$ (see Appendix E) are well recovered but the inhibitory $J^{i}$ is poorly recovered (inferred: $9.49\pm 3.69$; ground truth: $19.97$); each of the multiple initializations of neuLVM show convergence to the same result, suggesting that non-identifiability, more than the Baum-Viterbi algorithm, is the cause of partial parameter recovery; providing the ground truth latent population activities and taking longer recording times (40 seconds) improve parameter recovery. As discussed in Q4, applying a known external stimulus to the network could improve identifiability (in an informal sense).

It is very likely that, in most cases, neuLVM is non-identifiable. As for other mechanistic models of neuronal dynamics, a Bayesian estimation of the posterior distribution over the model parameters could remedy the problem of non-identifiability (Lueckmann et al., 2017; René et al., 2020). As mentioned in the Discussion, this natural next step is left for future work.

#### Q4: In the WTA experiment (Section 5.2), can the parametrization of connectivities $J$'s be relaxed?

In the WTA setup, removing the parametrization of the connectivities $J$'s (see Appendix E) is possible but it significantly decreases the performance of neuLVM. We hypothesize that this is due to the fact that the network spends most of its time around metastable fixed points, which do not provide the range of data necessary to constrain 9 arbitrary $J$'s (that is, $J$'s without the parametrization described in Appendix E). Consistent with this hypothesis, a preliminary follow-up experiment shows that, if the same network receives known external inputs perturbing the metastable states, the performance of neuLVM is restored. A thorough numerical investigation of the benefit of external perturbations (delivered via optogenetics for example) for neuLVM fitting is left for future work.


#### Q5: How could neuLVM be applied to real data?

We agree with the reviewers that, in the Discussion section, the question of how neuLVM could be applied to real data should be treated more thoroughly. Namely, we will clarify that neuLVM, on its own, cannot be reasonably applied to real data, without a systematic model comparison strategy (that is, many neuLVM's with different network architectures should be compared). To reduce the space of possible network architectures to be compared, clustering methods could be used to guide the coarse-graining of the observed neurons into homogeneous populations. Testing such an integrated approach on real data is beyond the scope of this work and we do not make any claim about the expected performance of this approach on real data.

---

### Meta-Review · Area_Chair_8EzT · 2022-08-26

**Recommendation:** Accept
**Confidence:** Certain

**Metareview:**

Dear authors,

Congratulations on your paper being accepted! The reviewers unanimously recommended acceptance. The reviewers made a number of recommendations on how to improve the paper further, in particular with respect to clarity of writing and explaining the motivation behind different analyses. We strongly encourage you use this feedback to improve the paper— if needs be additional clarifications can be added in the supplement. In addition, it would indeed be highly useful to make your source code publicly available, as you indicated in your response.

Best, your AC


**Award:**

No

---

### Decision · Program_Chairs · 2022-09-14

Accept